# Fungal-Modified Lignin-Enhanced Physicochemical Properties of Collagen-Based Composite Films

**DOI:** 10.3390/jof8121303

**Published:** 2022-12-16

**Authors:** Alitenai Tunuhe, Pengyang Liu, Mati Ullah, Su Sun, Hua Xie, Fuying Ma, Hongbo Yu, Yaxian Zhou, Shangxian Xie

**Affiliations:** 1Key Laboratory of Molecular Biophysics of MOE, Department of Biotechnology, College of Life Science and Technology, Huazhong University of Science and Technology, Wuhan 430074, China; 2College of Urban Construction, Wuchang Shouyi University, Wuhan 430074, China; 3Guangxi Shenguan Collagen Technology Research Institute, Guangxi Shenguan Collagen Biological Group, Wuzhou 543000, China

**Keywords:** collagen, lignin, fungal modification, composite films

## Abstract

Renewable and biodegradable materials have attracted broad attention as alternatives to existing conventional plastics, which have caused serious environmental problems. Collagen is a potential material for developing versatile film due to its biosafety, renewability, and biodegradability. However, it is still critical to overcome the low mechanical, antibacterial and antioxidant properties of the collagen film for food packaging applications. To address these limitations, we developed a new technology to prepare composite film by using collagen and fungal-modified APL (alkali pretreatment liquor). In this study, five edible and medical fungi, *Cunninghamella echinulata* FR3, *Pleurotus ostreatus* BP3, *Ganoderma lucidum* EN2, *Schizophyllum commune* DS1 and *Xylariaceae* sp. XY were used to modify the APL, and that showed that the modified APL significantly improved the mechanical, antibacterial and antioxidant properties of APL/Collagen composite films. Particularly, the APL modified by BP3, EN2 and XY showed preferable performance in enhancing the properties of the composite films. The tensile strength of the film was increased by 1.5-fold in the presence of the APL modified by EN2. To further understand the effect of fungal-biomodified APL on the properties of the composite films, a correlation analysis between the components of APL and the properties of composite films was conducted and indicated that the content of aromatic functional groups and lignin had a positive correlation with the enhanced mechanical and antioxidant properties of the composite films. In summary, composite films prepared from collagen and fungal biomodified APL showed elevated mechanical, antibacterial and antioxidant properties, and the herein-reported novel technology prospectively possesses great potential application in the food packaging industry.

## 1. Introduction

It is estimated that global plastic production has reached approximately 400 million metric tons in 2022, and these hard-to-degrade plastics pose a serious threat to the global environment and ecosystems [1,2]. Recent studies, in particular, have evidenced the bioaccumulation and biomagnification of microplastics will raise serious health issues for human beings [3]. Thus, most of the countries around the world have introduced a series of policy measures to control and even ban the usage of non-degradable plastics. Furthermore, numerous efforts are currently being made to improve the recycling process of plastics and to develop novel biodegradable materials to replace disposable plastics [4,5,6].

Biopolymers, which are mainly derived from natural materials, are considered to be the most promising packaging materials as alternatives to plastics as they are renewable and environmentally friendly [7,8]. Collagen, a major biopolymer component in animal skin, tendons and other various connective tissues, has been widely used as active food packaging material for its renewability, biocompatibility, biodegradability and protofibril-forming capabilities [9,10]. Collagen can form fibrils and networks through a self-assembly process with high surface uniformity and flexibility, but it is limited by weak antimicrobial and antioxidant properties, as well as low mechanical strength as a packaging material. To compensate for the performance deficiencies of collagen, many studies have been conducted by cross-linking the collagen with other natural polymers. Mehraj Ahmad et al. [11] developed a type of packaging film by laminating rice flour with fish gelatin, which showed significant UV absorption and improved thermal stability compared to gelatin film. Similarly, Emna Ben Slimane et al. [12] reported that the addition of chitosan could significantly enhance the UV-blocking properties and antioxidant activity of collagen film. Recently, some studies reported that lignin could be used to produce composite films with chitosan [13,14], gelatin [15] and polylactic acid [16,17] to enhance the antioxidant and antibacterial properties, as well as mechanical strength of the films. Such advantages of lignin as additives should be attributed to the enriched reactive functional groups (phenolic hydroxyl, carboxyl and methoxy groups, etc.) in lignin [18,19]. In addition, more than 300 billion tons of lignin are available in the biosphere, and numerous lignin waste is produced worldwide as by-products of the biorefinery and paper industry, while only a small fraction of lignin has been utilized [20]. Therefore, developing novel technologies to valorize lignin not only address the inadequacies of collagen films but also could be of great significance to the sustainability of the ecological environment.

Due to the heterogeneity and recalcitrance of lignin, the extraction and modification processes will have great effects on the resulting lignin structures and will lead to different properties of the corresponding composite materials. The main methods of lignin modification are chemical and biological treatments or their combinations [21,22]. Among them, biological modification is mild in conditions and seldom produces toxic and hazardous waste [22]. Our previous studies demonstrated that biomodified lignin could significantly promote the mechanical properties of composite materials (e.g., carbon fiber, asphalt binder) [23,24]. A recent study showed that white-rot fungal treatment can be used to modify the structure of lignin to be more favorable for the bio-based fiber materials [25]. Biomodification of lignin by fungi not only depolymerizes large molecules of lignin into smaller molecular units with more reactive groups but also increases the reactive functional groups of lignin [26]. Therefore, biomodified lignin can be used as a potential additive to improve the mechanical strength, antioxidant and antibacterial abilities of the collagen-based composite films.

In this study, alkali pretreatment liquor (APL), a kind of lignin-enriched residue produced from alkali pretreatment of lignocellulose, was used as one of the raw materials. Five edible and medicinal fungi were selected to biomodify the lignin in alkali pretreatment liquor (APL) based on our previous studies that showed different lignocellulosic degradation patterns of these fungi [27,28,29,30]. The biomodified APL with different fungi were further used to prepare composite films. The compositional changes of APL before and after fungal modification were investigated, as well as the mechanical, antibacterial and antioxidant properties of the produced biomodified APL-collagen composite films.

## 2. Materials and Methods

Fungal strains used in the study included *Cunninghamella echinulata* FR3, *Pleurotus ostreatus* BP3, *Ganoderma lucidum* EN2, *Schizophyllum commune* DS1 and *Xylariaceae* sp. XY, which were isolated and characterized in our previous studies [27,28,29,30].

The collagen used in this study was prepared by the Guangxi Shenguan Collagen Biological Group through industrial standard processes. Briefly, bovine sin was firstly pretreated by alkali, and eventually, collagen was obtained with enzyme extraction [31,32].

### 2.1. Pretreatment of APL

For the preparation of APL, 80 g corn stover (dry weight) was loaded into a 2 L conical flask at 10% (*w*/*w*) solid loading with 1% NaOH and pretreated for 1 h at 121 °C [33]. After pretreatment, 800 mL of distilled water was added, then filtered with 8 layers of gauze and centrifuged at 8000 rpm for 10 min to collect the supernatant. The pH was adjusted to 7 with 2 M HCl and sterilized at 121 °C for 20 min.

### 2.2. Biomodification of APL

The fungi were grown on potato-dextrose agar (PDA) slant at 28 °C for 7 days and were inoculated into potato dextrose broth (PDB) medium and cultured at 28 °C with shaking at 150 rpm for 7 days. A total of 10 mL of the pre-cultured fungi inoculum was transferred to 100 mL of sterilized APL in 250 mL Erlenmeyer flasks and incubated at 28 °C with 150 rpm of shaking for 7 days. The APL modified by *C. echinulate* FR3, *P. ostreatus* BP3, *G. lucidum* EN2, *S. commune* DS1 and *Xylariaceae* sp. XY were marked as AF, AB, AE, AD, and AX, respectively. The unmodified APL was marked as AC and taken as a control.

### 2.3. Component Changes before and after APL Biomodification

Total polysaccharide content of APL was measured by the anthrone sulfuric acid method [34]. The content of sugar with molecular weight less than 200 D in APL was collected by 200 D dialysis bags. The soluble solids content of APL was obtained by weighing 2 mL of the APL supernatant after drying at 60 °C to constant weight. Composition of soluble solids in APL were determined according to the “Determination of Extractives in Biomass” method from the National Renewable Energy Laboratory (NREL) [35].

### 2.4. Preparation of APL/Col Film

Collagen-based films were prepared by solution casting [36]. First, 6 mg/mL of collagen was directly dispersed in ultrapure water and stirred at low temperature until dispersed completely. Secondly, the precursor solution was prepared by adding modified APL at a dose of 2.8 mg/mL after the pH of collagen solution was up to 10, and 10% (*w*/*w*) glycerol was used as a plasticizer. Finally, 30 mL of precursor solution was poured into polyethylene plastic plates (90 × 90 mm) and dried in a constant temperature and humidity incubator (25 ± 0.5 °C, 52% RH) for 24 h. The collagen-based films with biomodified APL by different fungi were marked as C-AC (collagen with none-modified APL film), C-AF (collagen with *C. echinulate* FR3-modified APL film), C-AE (collagen with *G. lucidum* EN2-modified APL), C-AB (collagen with *P. ostreatus* BP3-modified APL film), C-AD (collagen with *S. commune* DS1-modified APL film) and C-AX (collagen with *Xylariaceae* sp. XY-modified APL film), based on aforementioned marks of fungi, respectively.

### 2.5. Characterization of APL/Col Film

#### 2.5.1. Morphology Analysis

All surfaces and cross sections of modified-APL by fungi composite with collagen films were observed using environmental scanning electron microscopy (ESEM) after being coated with a layer of gold [37]. The Fourier-transform infrared (FT-IR) spectra analysis of collagen-based films was determined by FT-IR (INVENIO-R, Bruker, Germany) with a spectral range of 400–4000 cm^−1^ at the attenuated total reflectance (ATR) mode. Then, X-ray diffractometer (XRD) patterns of film samples (2.0 × 6.0 cm) were performed on an X-ray diffractometer (x’pert3 powder, PANalytical B.V., Hollan). The data were recorded in the range (2θ) of 5°–80°, and the scanning rate was 16° min^−1^.

#### 2.5.2. Antibacterial Properties

To measure the antibacterial property of composite films, *Escherichia coli* (*E. coli*) ATCC 25922 was used to detect its growth incubated with or without the composite films [38]. *E. coli* was firstly inoculated in 10 mL of LB medium and cultured at 37 °C for 24 h before testing. Subsequently, normal saline was added to the samples to dilute it and fix the cell concentration to the range of 5–10 × 10^5^ CFU/ mL. Later, each film that had already been soaked in 10% NaCl solution was added into 3 mL of the test bacterial suspension, and the bacteria were cultured at 37 °C for another 48 h for testing the bacterial survival ratio according to the forming colonies on a plate.

#### 2.5.3. Antioxidant Properties and Water Contact Angle

The antioxidant property of the films was determined using ABTS methods [39]. Briefly, the antioxidant activity was measured by incubating small 0.2 g pieces of the film in the ABTS solution in the dark for 6 min. The absorbance after the reaction was obtained by a UV–visible spectrophotometer at 405 nm, while the ascorbic acid solution (70 μg/mL) was used as a positive control. The antioxidant capacity of the films was calculated according to the equation below:(1)$${\mathrm{ABTS}}^{+}\;\mathrm{scavenging}\;\mathrm{rate}\left(\%\right)=\frac{A_{0}-{\;A}_{1}}{A_{0}}\times 100\%$$
where A_0_ refers to the absorbance value of the control groups and A_1_ refers to the absorbance value of test groups.

The surface water contact angle of composite films was measured by a WCA analyzer by dropping 10 μL of ultrapure water on the surface of the films, followed by the prompt reading of the angle values on both sides of the drop [38].

#### 2.5.4. Transmittance, Film Thickness and Color Coordinates Analysis

The transmittance of composite films was determined using a UV–vis spectrophotometer (SolidSpec-3700, Shimadzu, Kyoto, Japan). Samples were cut into pieces with a size of 20 × 20 mm and put into test cells perpendicularly in front of the light beam before measuring. The film thicknesses of five randomly selected points on each film were read by a micrometer (Mitutoyo Spain, Elgoibar, Spain) [40]. The color parameters were determined by colorimeter (CIE). The composite film was placed on a white plate as a standard sample to measure the parameter L*(luminance (range from 0 (black) to 100 (white)), a* (colorimetric parameters (range from −80 (green) to 100 (red)) and b* (colorimetric parameters (range from −80 (blue) to 70 (yellow)). The color change ΔE was calculated using the equation (Equation (2)) given below [40].
(2)$${\Delta E}^{*}={\left[{\left(L_{0}^{*}-L^{*}\right)}^{2}+{\left(a_{0}^{*}-a^{*}\right)}^{2}+{\left(b_{0}^{*}-b^{*}\right)}^{2}\right]}^{1/2}$$

L_0_, a_0_ and b_0_ refer to the color parameters of a white standard sample. Whereas L, a, and b values refer to the color parameters of the samples. Three measurements were tested for each sample.

#### 2.5.5. Mechanical Strength

Mechanical properties, including tensile strength (TS) and elongation at break (EAB), were obtained by using an electronic universal testing machine (ACS-J, Shimadzu, Japan). Three parallel samples (10 × 60 mm) of each formulation were cut and fixed between handles and stretched at 5 mm/min. The average of mechanical properties was calculated after measuring three replicate samples values [41,42].

#### 2.5.6. ^31^P NMR Analysis

The diverse hydroxyl groups in different fungi-modified lignin fractions were determined by 31P NMR (nuclear magnetic resonance). Approximately 30 mg of APL was dissolved into 500 μL solution A, which is constituted by pyridine and deuterated chloroform (1.6:1, *v*/*v*). Secondly, 100 μL 0.1 mM N-Hydroxynaphthalimide and 100 μL 0.0143 M chromium (III) acetylacetanoate were used as the internal standard and relaxation agent, respectively. Finally, 100 μL of phosphitylating reagent II (2-chloro-4,4,5,5-tetramethyl-1,3,2-dixaphospholane) was added and the mixture was left at room temperature for 20 min with continuous ultrasonic dissolution. The spectra were acquired using a Bruker 600 MHz spectrometer equipped with a Quad probe dedicated to ^31^P acquisition. The spectra were accumulated with a time delay of 25 s between pulses [43].

#### 2.5.7. GC-MS Analysis of Compounds in APL

Aromatic compounds in APL supernatant were detected by GC-MS (gas chromatography/mass spectrometry, Thermo Fisher Scientific, Austin, TX, USA) according to the previous method [44]. At first, the supernatant was obtained by uninoculated and fungal inoculated APL samples (10 mL), centrifuged (8000 rpm for 10 min), and acidified to a pH of 1–2 with concentrated HCl mixed with Ethylvanillin (final concentration 0.25 mg/mL) as an internal standard. After that, three times of volume of ethyl acetate was added to shake for 4 h to extract completely. Then, the organic layer was collected and dried with anhydrous Na_2_SO_4_ before it was filtered through filter paper. Subsequently drying under a stream of nitrogen, 400 μL dioxane and 10 μL pyridine were added to samples followed by 200 μL BSTFA. Finally, the mixture was heated for a silylation reaction at 60 °C for 30 min.

Silylated compounds (5 μL) were injected directly into the GC-MS, and the split ratio was 5:1. Agilent HP-5MS cap (30 m × 0.25 mm inner diameter, and 0.25 μm membrane thick) was used as the analysis column, and the nitrogen was used as carrier gas (1 mL/min) respectively. The column temperature program was initially maintained at 50 °C for 3 min while increased at 8 °C/min to 140 °C and maintained for 5 min. After that, the column temperature was increased at 4 °C/min to 170 °C and maintained for 5 min and sequentially increased at 10 °C/min to 300 °C and kept for 5 min. The temperature of the transmission line and ion source was 150 and 230 °C, and the data collection was delayed for 3 min. The electronic ionization (EI) mass spectrum within the range of 50–500 (*m*/*z*) was recorded in the full scan mode under the electronic energy of 70 EV. Eventually, the corresponding compound was acquired by comparing the obtained mass spectrum with the NIST05 library and then normalizing the acquired peak height against that of an internal standard.

## 3. Results and Discussion

### 3.1. Fungal Biomodification of APL

APL usually contains polysaccharides and monosaccharides due to alkali pretreatment, and the content of these saccharides will have different impacts on the mechanical properties of the subsequent films. As shown in Figure 1A, the total sugar in APL decreased after biological modification, indicating that fungi can utilize the sugars in APL. The sugar with low molecular weight (<200D) was more preferred by all the five fungi compared to high molecular weight sugar (>200D). Xylariaceae sp. XY showed the most efficient saccharides utilization among the five fungi, which consumed 55.2% of the total sugar. Meanwhile, the APL treated by fungi *P. ostreatus* BP3, *S. commune* DS1 and *Xylariaceae* sp. XY showed a significant reduction in soluble solid content. The components analysis further confirmed the APL degradation by the fungi. The lignin content in APL treated by *P. ostreatus* BP3, *S. commune* DS1 and *Xylariaceae* sp. XY was significantly increased compared to original APL, while the cellulose and hemicellulose content in the APL treated by all the fungi were decreased due to the consumption of sugars by the fungi (Table 1). The component changes of the APL treated by different fungi could cause different material properties of the collagen-APL films.

### 3.2. The Characterization of APL/Col Films

#### 3.2.1. SEM Analysis

It is straightforward to illustrate the features of the surface and cross sections of the composite films by SEM analysis [36,37]. As shown in Figure 2, the pure collagen film (A) showed a rough surface with clearly visible protrusions, while the addition of APL could obviously decrease the protrusions and make the surface of collagen-APL films much smoother. In addition, the cross sections of composite films (Figure 2b–g) were more compact than the collagen film (Figure 2a), and the fiber arrangement in cross sections of the composite film with fungi-modified APL (Figure 2c–g) were more orderly than that film with unmodified APL (Figure 2b). This might be caused by a formed hydrogen bond between collagen polypeptide chains and the hydroxide radical of APL, which was significantly increased after fungal modification [45]. Hence, the surface irregularity could be significantly changed after adding APL to the collagen film.

#### 3.2.2. ATR-FTIR and XRD Analysis

ATR-FTIR spectra of APL and collagen-based films are shown in Figure 3A–C. Firstly, the infrared adsorption wavenumber of characteristic peaks of lignin displayed from 500 to 1600 cm^−1^, and the adsorption peaks at 1600, 1517, and 1461 cm^−1^ were an aromatic ring stretching mode of the phenyl-propane skeleton, which is shown in all spectra of Figure 3B, indicating that all samples shared the basic skeleton of lignin, while the bands observed at 1166 and 839 cm^−1^ were C=O vibration bands in conjugated ester groups (typical for GSH lignin) and aromatic C-H out of plain vibrations in all positions of H units, respectively [46]. As shown in Figure 3C, the infrared adsorption of different combined collagen films maintained the characteristic peaks of amide A and B, amide I, II and III of collagen. The peaks at 3301 and 2930 cm^−1^ were amide A and B, respectively, mainly due to the N-H stretch coupled with hydrogen bonding [40]. The bands at 1642, 1548 and 1237 cm^−1^ represented amide I (stretching vibrations of C-O groups), II (N-H bending and C-N stretching) and III (the vibrations in C-N and N-H groups of bound amides), respectively. These observed characteristic peaks of APL/Col composite films were consistent with other research on FTIR results of collagen fiber [37,38,40,41], which indicated that collagen films with modified APL could well maintain the triple-helical structure of collagen.

To further confirm the containing of collagen with a triple helix, the films were analyzed by X-ray diffraction (Figure 3D). The peak at 2θ = 8.1° was ascribed as the collagen chain, and the intensity of 8.1° indicated the intermolecular lateral packing distance between the collagen molecular chains [47]. The peak at 2θ = 21° was related to the distance between amino acid residues (0.44 nm) along the helix of collagen [47]. In this study, the films compositing with APL showed no obvious shift in the aforementioned peaks, demonstrating the stability of the triple helix structure. However, the intensity of the peak at 2θ = 8.1° in the collagen compositing with DS1-modified APL was significantly decreased compared to that in the other films, indicating the declination of the triple helix structure in the film in the presence of APL modified by DS1.

#### 3.2.3. Antibacterial Properties

Antibacterial packaging film is one of the promising technologies for food preservation [48]. The antibacterial property of collagen-based films was determined against *E. coli* (Figure 4). The results showed that there was no obvious inhibition to the colony growth of *E. coli* after incubating with the collagen film (Figure 4B) and composite film with unmodified APL (Figure 4C), respectively. Interestingly, all the collagen composite films with fungal-modified APL showed obvious antibacterial activity (Figure 4D–H), and among which the APL modified by *P. ostreatus* BP3, *S. commun* DS1 and *Xylariaceae* sp. XY showed the strongest antibacterial activity to *E. coli* (Figure 4F–H). Combined with the APL composition analysis (Table 1), it showed that the higher content of lignin in the fungal-modified APL exhibited stronger antibacterial corresponding collagen composite films. These results suggested that the antibacterial activity may relate to the content and the structure of fungal-modified lignin in the composite films [49].

#### 3.2.4. Antioxidant Activities and Water Resistance

Antioxidant activity of packaging film plays an essential role in maintaining the freshness of food. To further investigate the effect of fungal-modified APL on the antioxidant property of APL/Col films, the antioxidant activity of the prepared films was measured by ABTS assays [50]. As shown in Figure 5A, collagen films without APL showed a low ABTS scavenging rate, which is consistent with previous studies that show that the collagen has some antioxidant activity [42]. However, all the composite films with APL, no matter the unmodified or modified by fungi, displayed superior antioxidant properties compared to the collagen films. This suggested that the antioxidant property of the films was mainly contributed by lignin, which has been approved with impressive antioxidant capability [49].

Water contact angle (WCA) is a basic index to characterize the wettability of a food packaging material. The WCA of the collagen film was 96.97° (Figure 5B), demonstrating that the collagen film was hydrophobic [51]. The addition of APL reduced the WCA of the composite films, while the composite films with fungal-modified APL showed higher WCA compared to that with unmodified APL. Considering the component difference in the APL before and after fungal modification, the changes of WCA with different APL could be caused by the containment of polysaccharides, which contained a large number of hydroxyl groups forming hydrogen bonds with water to increase the hydrophilia [52]. Among the five fungal-modified APL, the APL modified by *C. echinulate* FR3 and *S. commun* DS1 showed substantial improvement compared to the WCA of the composite films, which possibly will be due to the maximum sugar utilization by *C. echinulate* FR3 and *S. commun* DS1 in the APL fermentation processes.

#### 3.2.5. Transmittance and Color Coordinates

To further investigate the impact of APL on the morphology of composite films, the transmittance and color coordinates of the prepared composite films were measured. The films were all translucent and homogeneous, while the films with APL exhibited yellow due to the introduction of color groups from APL (Figure 6A). The transparency of the films was measured by UV-vis spectrophotometer (300–800 nm). The addition of APL significantly reduced the light transparency of the composite films compared to the collagen film, where the transparency of collagen was 89.58% at 600 nm, and that of the composite film with APL was decreased to 58.18–68.38%. The decrease in such transparency should be caused by the enriched chromophores in lignin [40]. However, the transparency of the composite films at the ultraviolet spectrum (300–380 nm) displayed totally different properties compared to the collagen film. The composite films totally blocked the transparency of ultraviolet light, while the collagen film showed high ultraviolet light transparency (68% up to 86% at 300 to 380 nm). This result suggested that the composite films had strong UV-resistant capacity contributed by the addition of lignin [53].

The colorimetric parameter analysis further confirmed the changes in color morphology caused by the addition of APL to the composite films (Table 2). The luminance of the films was significantly decreased in the presence of APL. For instance, the luminance (indicated by L value) of the collagen film was 96, while the L value of the composite film with unmodified APL was decreased to 64.72, and the L value continued to decrease to 47.6–55.2 after compositing with fungal-modified APL. The colorimetric parameter b, indicating the color change from blue to yellow, was consistent with the digital photo, as shown in Figure 6A, where the composite film with APL is yellower than the collagen film. It was also noticed that parameter b of the composite films with fungal-modified APL was reduced compared to that with unmodified APL, indicating some chromophoric groups of APL may be degraded during fungal fermentation [54].

#### 3.2.6. Mechanical Properties

The mechanical resistance and ductility are essential properties used to evaluate the function and application of the film. The tensile strength (TS) and elongation at break (EAB) of the composite films were analyzed to index the mechanical properties of the films (Figure 7). Different APL showed different impacts on the TS and EAB of the composite films. The addition of unmodified APL showed no significant change to the TS but enhanced EAB relative to its corresponding composite film, while the APL modified by fungi *Ganoderma lucidum* EN2, *Pleurotus.* sp. BP3 and *Xylariaceae* sp. XY significantly enhanced both TS and EAB of their corresponding composite films. For example, the tensile strength of the composite film with EN2-modified APL was enhanced to 51 MPa, which is 50% higher than the collagen-based control film. These results demonstrated our hypothesis that the APL modification with fungi would affect the mechanical properties of the composite film, as its components and structure change during fungal fermentation. Since various fungi showed different lignin degradation capacities and biomass degradation patterns, the modified APL showed exceptional differences in their components and structure. However, it is still not clear how the structural components, interunit linkages and functional groups of lignin impact the properties of composite films, and thus, this needs to be studied further systematically.

#### 3.2.7. Correlation Analysis between the APL Components and Film Properties

Even though the mechanism of the APL impacting the properties of composite films is unclearly revealed in the current study. It is still very important to investigate if there were correlations between the APL components and the film properties, which could guide a further in-depth fundamental study and design the applicable composite films with different properties based on the diverse fungal-biomodified APL. To conduct the correlation analysis, the contents of major components (lignin, cellulose and hemicellulose, Table 1), different functional groups measured by NMR (nuclear magnetic resonance, Appendix A), and soluble total phenols and acids in different types of APL (Appendix A) were used to analyze their correlations with the antioxidant, hydrophobic (WCA, Figure 5B) and mechanical properties (EAB and TS, Figure 8) of the composite films. The results showed that the content of aromatic compounds and aromatic functional groups in APL was positively correlated with the antioxidant capability of the composite film, while it seemed that the content of the acid-soluble lignin in APL showed a negative correlation with the antioxidant capability of the composite films. For the hydrophobicity indexed by the WCA of the films, it seemed that the content of the lignin had a positive correlation with WCA and the content of phenolic hydroxyl in the lignin had a negative correlation with it. The content of the cellulose in APL showed a negative correlation with the mechanical properties (both TS and EAB) of the composite films, and the content of hemicellulose in APL showed a positive correlation with the mechanical properties. In addition, the content of lignin showed a positive correlation with EAB, and the content of hydroxyl groups showed a negative correlation with EAB. It is notable here that the cellulose detected in APL was soluble with small molecules released from lignocellulose after alkali treatment [55]. Previous studies also revealed that the high content of soluble sugar decreased the mechanical properties of collagen. Therefore, the fungi, which could efficiently utilize the small molecular sugar, can significantly increase the content of the lignin by consuming the soluble cellulose, while most of the hemicellulose in APL is cross-linked with lignin to form a lignin–carbohydrate complex [56], which could enhance the mechanical properties. Even though these results showed some correlations between the APL components and the film characterization, an in-depth study still needs to be conducted to reveal the mechanisms of the APL components affecting each property of the composite films.

## 4. Conclusions

In conclusion, an active packaging film was successfully prepared with collagen and fungal-modified APL. In this study, five edible and medicinal fungi, *Cunninghamella echinulata* FR3, *Pleurotus ostreatu* BP3, *Ganoderma lucidum* EN2, *Schizophyllum commune* DS1 and *Xylariaceae* sp. XY, were used to modify the APL and significantly change the components and structures of APL, which were demonstrated by composition analysis, GC-MS and NMR analysis. The different APL modified by different fungi showed various effects on the morphology and properties of their corresponding composite films with collagen. Overall, the APL biomodified by fungi significantly enhanced the antibacterial, antioxidant, hydrophobic and mechanical properties of the composite films at different levels. Among which, the APL modified by fungi *P. ostreatu* BP3 and *Xylariaceae* sp. XY showed the most significant improvement in the mechanical properties, as well as antibacterial and antioxidant activities. A correlation analysis was conducted between the APL components and the properties of the composite films, which could provide further guidance for understanding the mechanism in detail and may provide more insight into the application design for future development. This study first demonstrated that the biomodified lignin residue by fungi could be a potential ingredient to produce biodegradable and functional composite films with collagen for large-scale applications in food packaging.

## 5. Patent

Application number 202211609643.7.

## Figures and Tables

**Figure 1 jof-08-01303-f001:**
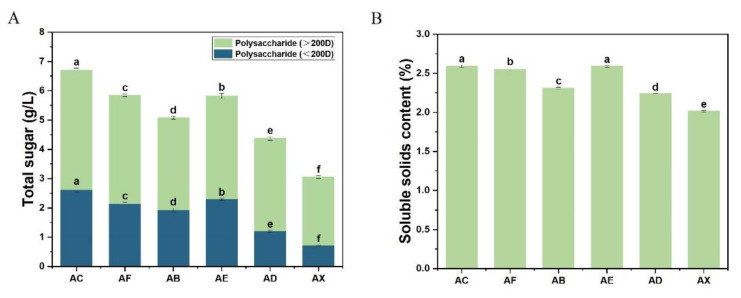
Contents of sugar with different molecular weights (**A**) and contents of soluble solids (**B**) in APL after fungal biomodification. Polysaccharides of different molecular weights were graded through a 200D dialysis bag. AC (none-modified APL); AF (*C. echinulate* FR3-modified APL); AB (*P. ostreatus* BP3-modified APL); AE (*G. lucidum* EN2-modified APL); AD (*S. commune* DS1-modified APL); AX (*Xylariaceae* sp. XY-modified APL). For the comparison between different biological modifications, different lowercase letters represent significant differences (*p* < 0.05).

**Figure 2 jof-08-01303-f002:**
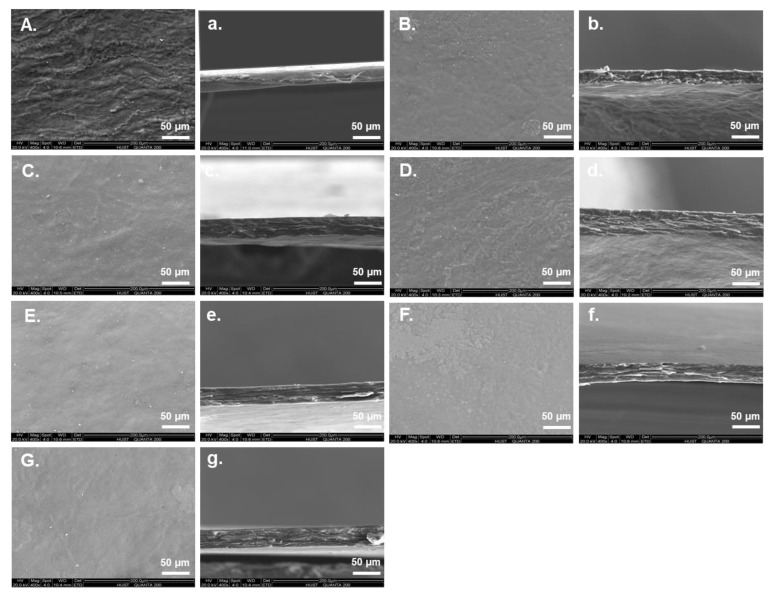
The SEM images (400×) of the collagen film surfaces (**A**–**G**) and cross-sections (**a**–**g**). **A/a**: Control (collagen film); **B/b**: collagen with none-modified APL film; **C/c**: collagen with *C. echinulate* FR3-modified APL film; **D/d**: collagen with *G. lucidum* EN2-modified APL film; **E/e**: collagen with *P. ostreatus* BP3-modified APL film; **F/f**: collagen with *S. commune* DS1-modified APL film; **G/g**: collagen with *Xylariaceae* sp. XY-modified APL film.

**Figure 3 jof-08-01303-f003:**
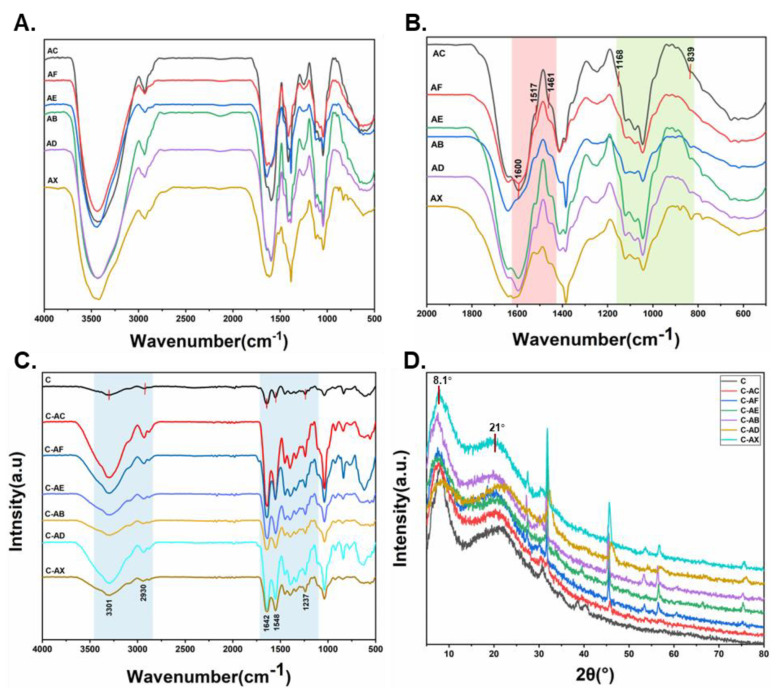
ATR−FTIR spectrum (**A**–**C**) and X-ray diffraction spectrum (**D**) of APL with collagen films. (**A**): wavenumber from 500 to 4000 cm^−1^ of the different fungi-modified APL; (**B**): wavenumber from 500 to 2000 cm^−1^of the different fungi-modified APL; (**C**): wavenumber from 500 to 4000 cm^−1^ of the different fungi-modified APL with collagen film. C (collagen film); C−AC (collagen with none−modified APL film); C−AF (collagen with *C. echinulate* FR3−modified APL film); C−AE (collagen with *G. lucidum* EN2−modified APL film); C−AB (collagen with *P. ostreatus* BP3−modified APL film); C−AD (collagen with *S. commune* DS1−modified APL film); C−AX (collagen with *Xylariaceae* sp. XY−modified APL film).

**Figure 4 jof-08-01303-f004:**
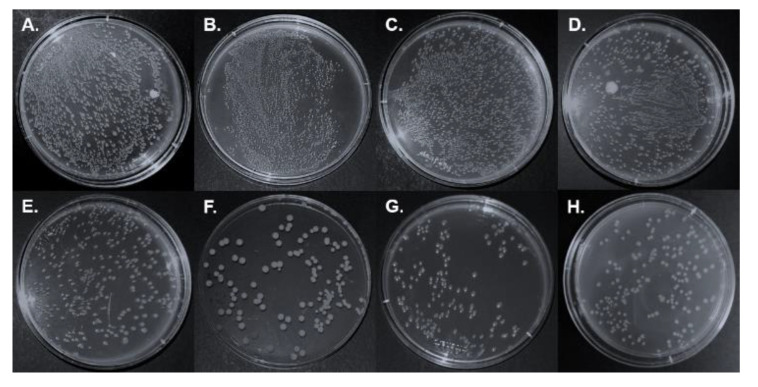
Different films co-cultured with *E. coli* after 48h and observation of effects on the growth inhibition of *E. coli*. (**A**) *E. coli*; (**B**) collagen film; (**C**) collagen with none-modified APL film; (**D**) collagen with *C. echinulate* FR3-modified APL film; (**E**) collagen with *G. lucidum* EN2-modified APL film; (**F**) collagen with *P. ostreatus* BP3-modified APL film; (**G**) collagen with *S. commune* DS1-modified APL film; (**H**) collagen with *Xylariaceae* sp. XY-modified APL film.

**Figure 5 jof-08-01303-f005:**
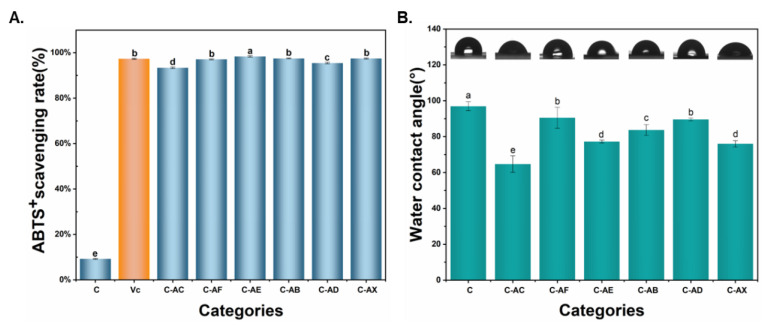
The ABTS+ scavenging rate (**A**) and WCA (**B**) of the collagen-based films. C (collagen film); C-AC (collagen with none-modified APL film); C-AF (collagen with *C. echinulate* FR3-modified APL film); C-AE (collagen with *G. lucidum* EN2-modified APL film); C-AB (collagen with *P. ostreatus* BP3-modified APL film); C-AD (collagen with *S. commune* DS1-modified APL film); C-AX (collagen with *Xylariaceae* sp. XY-modified APL film). For the comparison between different biological modifications, different lowercase letters represent significant differences (*p* < 0.05).

**Figure 6 jof-08-01303-f006:**
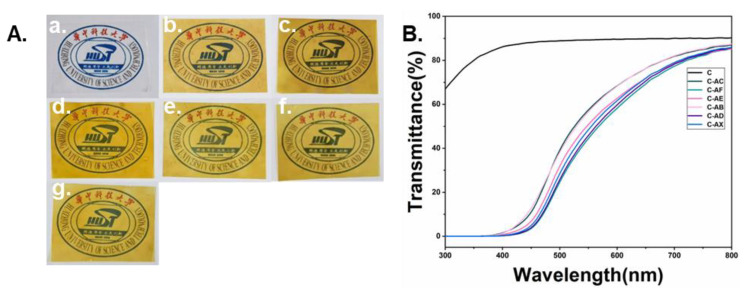
The digital photos (**A**) and the transmittances (**B**) of the collagen-based films. (**a**): collagen film; (**b**): collagen with non-modified APL film; (**c**): collagen with *C. echinulate* FR3-modified APL film; (**d**): collagen with *G. lucidum* EN2-modified APL film; (**e**): collagen with *P. ostreatus* BP3-modified APL film; (**f**): collagen with *S. commune* DS1-modified APL film; (**g**): collagen with *Xylariaceae* sp. XY-modified APL film.

**Figure 7 jof-08-01303-f007:**
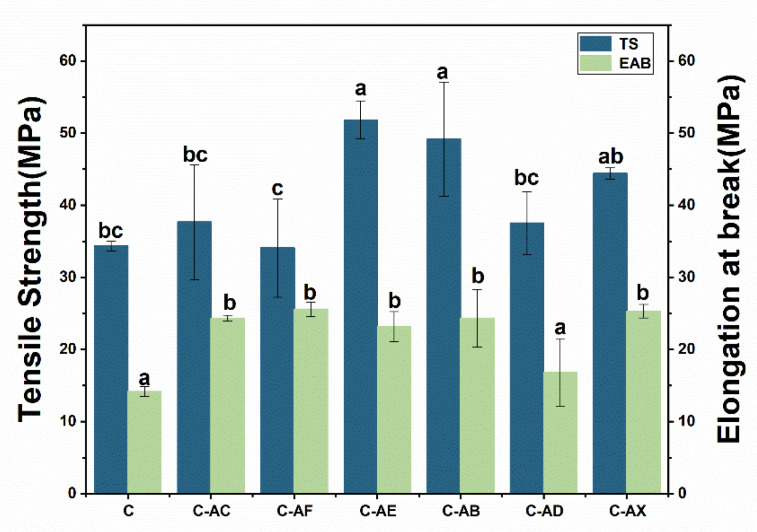
Mechanical properties of collagen-based film. TS = tensile strength, EAB = elongation at break. C (collagen film); C-AC (collagen with none-modified APL film); C-AF (collagen with *C. echinulate* FR3-modified APL film); C-AE (collagen with *G. lucidum* EN2-modified APL film); C-AB (collagen with *P. ostreatus* BP3-modified APL film); C-AD (collagen with *S. commune* DS1-modified APL film); C-AX (collagen with *Xylariaceae* sp. XY-modified APL film). For the comparison between different biological modifications, different lowercase letters represent significant differences (*p* < 0.05).

**Figure 8 jof-08-01303-f008:**
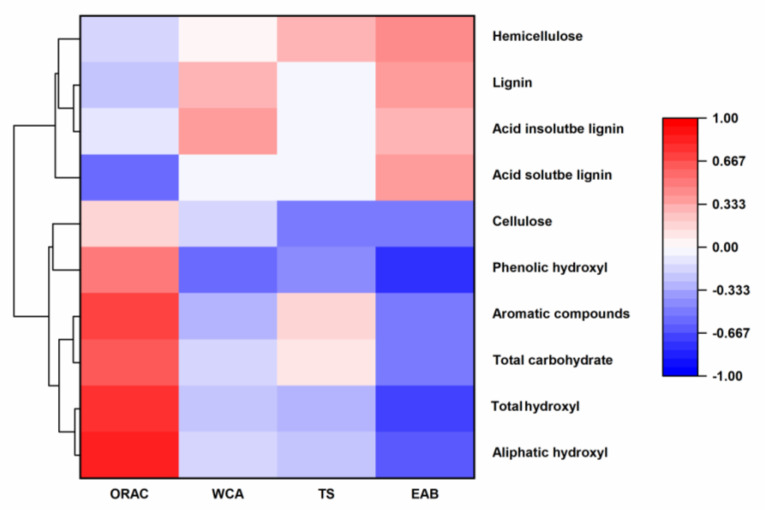
The heatmap combined with the Pearson correlation coefficient among the different factors. Factors including hemicellulose, lignin, acid-soluble lignin, acid insoluble lignin and cellulose from soluble solids; phenolic hydroxyl, total carbohydrate, total hydroxyl and aliphatic hydroxyl by 31P NMR and aromatic compounds by GC−MS were related to the antioxidant properties (ORAC), hydrophobicity (WCA), and mechanical properties (TS and EAB). Different color blocks represent different correlation coefficient values of the corresponding variables, using the color scale as a reference. ORAC (antioxidant activity obtained by measuring ABTS^+^ clearance), WCA (water contact angle), TS (tensile strength), EAB (elongation at break).

**Table 1 jof-08-01303-t001:** Soluble solids components of unmodified and biomodified APL.

	Ash%	Lignin%	Cellulose%	Hemicellulose%	Other%
AC	0.1 ± 0.02 ^bcd^	18.61 ± 0.24 ^c^	27.28 ± 0.78 ^a^	51.85 ± 0.94 ^a^	2.16 ± 1.46 ^d^
AF	0.22 ± 0.05 ^a^	18.62 ± 0.33 ^c^	25.85 ± 0.21 ^b^	48.48 ± 1.23 ^b^	6.82 ± 1.49 ^c^
AE	0.14 ± 0.07 ^ab^	18.39 ± 0.13 ^c^	23.71 ± 0.69 ^b^	44.55 ± 0.37 ^c^	5.78 ± 0.74 ^c^
AB	0.04 ± 0 ^cd^	19.93 ± 0.34 ^b^	23.96 ± 0.42 ^b^	50.28 ± 0.43 ^a^	13.21 ± 0.94 ^a^
AD	0.13 ± 0.04 ^abc^	21.17 ± 0.57 ^a^	22.01 ± 0.5 ^d^	46.92 ± 1.01 ^b^	9.76 ± 2.03 ^b^
AX	0.03 ± 0.02 ^d^	20.75 ± 0.48 ^ab^	23.73 ± 0.13 ^c^	50.46 ± 0.28 ^a^	5.03 ± 0.64 ^c^

AC (none-modified APL); AF (*C. echinulate* FR3-modified APL); AB (*P. ostreatus* BP3-modified APL); AE (*G. lucidum* EN2-modified APL); AD (*S. commune* DS1-modified APL); AX (*Xylariaceae* sp. XY-modified APL). Data in the same column followed by different lowercase letters are significantly different (*p* < 0.05).

**Table 2 jof-08-01303-t002:** The thickness and color parameters of the collagen-based films.

	Thickness	L	a	b	ΔE
C	42.2 ± 3.11 ^b^	96 ± 0.04 ^a^	−1 ± 0.04 ^g^	2.43 ± 0.02 ^g^	2.13 ± 0.02 ^g^
C-AC	52.6 ± 3.65 ^a^	64.72 ± 0.01 ^b^	15.91 ± 0.02 ^d^	58.04 ± 0.03 ^a^	67.64 ± 0.03 ^a^
C-AF	48.8 ± 5.02 ^a^	51.91 ± 0.02 ^d^	15.14 ± 0.01 ^f^	40.82 ± 0.01 ^c^	61.97 ± 0.01 ^c^
C-AE	49.6 ± 4.67 ^a^	48.62 ± 0.04 ^e^	16.33 ± 0.04 ^c^	36.16 ± 0.01 ^d^	61.86 ± 0.04 ^d^
C-AB	50.4 ± 2.7 ^a^	55.2 ± 0.01 ^c^	15.84 ± 0.02 ^e^	45.44 ± 0.03 ^b^	63.05 ± 0.02 ^b^
C-AD	51.8 ± 0.84 ^a^	47.6 ± 0.01 ^f^	16.4 ± 0.01 ^b^	34.36 ± 0.03 ^e^	61.67 ± 0.01 ^e^
C-AX	49.6 ± 7.77 ^a^	47.11 ± 0.01 ^g^	16.54 ± 0.02 ^a^	31.93 ± 0.03 ^f^	60.8 ± 0.02 ^f^

L (luminance (range from 0 (black) to 100 (white)), a (colorimetric parameters (range from −80 (green) to 100 (red)), b (colorimetric parameters (range from −80 (blue) to 70 (yellow)). Data in the same column followed by different lowercase letters are significantly different (*p* < 0.05). C (collagen film); C-AC (collagen with none-modified APL film); C-AF (collagen with *C. echinulate* FR3-modified APL film); C-AE (collagen with *G. lucidum* EN2-modified APL film); C-AB (collagen with *P. ostreatus* BP3-modified APL film); C-AD (collagen with *S. commune* DS1-modified APL film); C-AX (collagen with *Xylariaceae* sp. XY-modified APL film).

## Data Availability

Not applicable.

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
