# Peer review of "Fungal-Modified Lignin-Enhanced Physicochemical Properties of Collagen-Based Composite Films"

_jof, 2022, doi:10.3390/jof8121303_

Round 1

Reviewer 1 Report

In this interesting article, Tunuhe et al., show how fungi can be used to modify or repurpose alkaline pretreatment liquor waste, which can then produce APL/collagen films with enhanced properties that are valuable in food packing. My only comments for improvement are very minor: (1) consider expanding on APL origin, issues and the need for its modification.; (2) the manuscript has multiple grammar errors (3) Some figures (like Figure 2) could be bigger. 

Author Response

1.Consider expanding on APL origin, issues and the need for its modification.

The origin of APL was added at line 83-84. The heterogeneity and recalcitrance of lignin will affect the properties of the final material as we stated in line 69, therefore, it is needed to modify the APL to enhance its properties as discussed in line 70-81.

  1. The manuscript has multiple grammar errors.

We tried our best to improve the manuscript and have revised the manuscript. These changes will not influence the content and framework of the paper. And here we did not list the changes but marked in yellow in the revised paper. We appreciate for Editors/Reviewers’ warm work earnestly.

  1. Some figures (like Figure 2) could be bigger. 

 We deeply appreciate the reviewer’s suggestion. According to the reviewer’s comment, we adjusted the size of Figure 2, Figure 3, Figure 5 and Figure 6.

Reviewer 2 Report

The manuscript describes the feasibility of using fungal utilization of alkaline pretreated liquor as a way to valorize lignin waste through the production of collagen-based composite films. The authors compare the effect five edible medical fungi on the morphology and properties of the composite films prepared, obtaining improvement in mechanical, antibacterial and antioxidant properties. Authors present the correlation analysis between the APL components and film properties. And as they state, an in-depth study is needed to understand the mechanism that leads to the improvement of film properties. However, in my opinion the subject of the paper is of interest, it is correctly written  and it can be published after some very  minor corrections that I list above:

Line 93. Some reference on the previous studies where the fungi were isolated and characterized, should be given.

Line 96. Also some reference on the process used to obtain the collagen should be added.

Author Response

  1. Line 93. Some reference on the previous studies where the fungi were isolated and characterized, should be given.

Thank you for the suggestion. The references on the previous studies of the fungi were added as shown in line 94. 

  1. Line 96. Also some reference on the process used to obtain the collagen should be added.

We sincerely appreciate the valuable comments. We have checked the literature carefully and added more references on Line 96 about the way of collagen extraction in the revised manuscript.
